# DNA Accounting: Tallying Genomes to Detect Adulterated Saffron

**DOI:** 10.3390/foods10112670

**Published:** 2021-11-03

**Authors:** Antoon Lievens, Valentina Paracchini, Danilo Pietretti, Linda Garlant, Alain Maquet, Franz Ulberth

**Affiliations:** 1European Commission, Joint Research Centre, B-2440 Geel, Belgium; danilo.pietretti@ec.europa.eu (D.P.); linda.garlant@ec.europa.eu (L.G.); alain.maquet@ec.europa.eu (A.M.); franz.ulberth@ec.europa.eu (F.U.); 2European Commission, Joint Research Centre, I-21027 Ispra, Italy; valentina.paracchini@ec.europa.eu

**Keywords:** food fraud, saffron, digital PCR, next generation sequencing

## Abstract

The EU General Food Law not only aims at ensuring food safety but also to ‘prevent fraudulent or deceptive practices; the adulteration of food; and any other practices which may mislead the consumer’. Especially the partial or complete, deliberate, and intentional substitution of valuable ingredients (e.g., Saffron) for less valuable ones is of concern. Due to the variety of products on the market an approach to detect food adulteration that works well for one species may not be easily applicable to another. Here we present a broadly applicable approach for the detection of substitution of biological materials based on digital PCR. By simultaneously measuring and forecasting the number of genome copies in a sample, fraud is detectable as a discrepancy between these two values. Apart from the choice of target gene, the procedure is identical across all species. It is scalable, rapid, and has a high dynamic range. We provide proof of concept by presenting the analysis of 141 samples of Saffron (*Crocus sativus*) from across the European market by DNA accounting and the verification of these results by NGS analysis.

## 1. Introduction

Adulteration and mislabeling of food has been known since biblical times and probably goes back as far as when food started to be traded [1]. Unsurprisingly, laws and regulations to combat these practices can also be found throughout history: ancient food regulations are referred to in Egyptian, Chinese, Hindu, Greek, and Roman texts. Today, the EU General Food Law—next to protecting public health—aims to ‘prevent fraudulent or deceptive practices; the adulteration of food; and any other practices which may mislead the consumer’ (Article 8 of Regulation (EC) No 178/2002 ). A commonly accepted definition is that ‘Food fraud includes adulteration, deliberate and intentional substitution, dilution, simulation, tampering, counterfeiting, or misrepresentation of food, food ingredients, or food packaging; or false or misleading statements made about a product for economic gain’ [2]. Although food fraud is not explicitly defined in EU legislation, the Food Information to Consumers Regulation (Regulation (EU) No 1169/2011) concretises the relevant aspects of the General Food Law in Article 7 (Fair information practices) by stipulating that food information shall not be misleading, particularly as to the characteristics of the food and, in particular, as to its nature, identity, properties, composition, quantity, durability, country of origin or place of provenance, and method of manufacture or production.

Herbs and spices are among the group of ingredients most vulnerable to fraudulent manipulation, mostly due to their high economic value, which makes them an attractive target. Furthermore, international trade and highly complex supply chains, involving various actors and processing steps, contribute to the presence of adulterated herbs and spices on the market [3,4]. Macro- and microscopic examination of morphological features, physico-chemical methods (e.g., ash, volatile oil content), and determination of aromatic principles typical for the product (e.g., members of the terpenoid or vanillylamide families) are traditionally used for quality grading and authentication of herbs and spices.

More advanced analytical methods, often in combination with machine learning algorithms, use proteins, metabolites, or DNA, either in a targeted or untargeted manner, to authenticate plant material [5]. Particularly, DNA based methods have equipped forensic analysts with highly specific, sensitive, and cost-effective tools because the genetic makeup of their targets is not influenced by environmental or physiological factors. The application of DNA technology for food authentication [6,7,8] and specifically for plant material [9,10] has been extensively reviewed. They make use of DNA polymorphism between species and most of them include a polymerase chain reaction (PCR) to amplify DNA. Species-specific PCRs targeting a product specific nucleotide sequence are very attractive because of their sensitivity, specificity, reproducibility, and ability to detect low target amounts. They are particularly useful to verify the claimed identity of the product or to detect the presence of another non-declared species. This approach is less appropriate if the identity of the adulterant(s) is not known, although multiplexed assays can provide a solution as long as the number of potential adulterants is small. Non-targeted methods, on the other hand, are able to find a wider range of biological contaminants. In most such methods a ‘fingerprint’ is generated by targeting variable length sequences which is then used to confirm purity compared to a database to find contaminants [11,12].

DNA barcoding has gained popularity for identifying animal as well as plant species [13,14]. Again, the technique works well if applied to single species; if the sample contains more than one species, the barcoding regions from all of them can be amplified resulting in difficult-to-interpret sequences in the Sanger sequencing step. Meta-barcoding using Next Generation Sequencing (NGS), a high-throughput, parallel, sequencing technology, offers a solution to this problem. Although the technique is becoming more widely available, it requires access to dedicated instrumentation, bioinformatics pipelines, and experienced operators.

We have developed a method called ‘DNA accounting’ for screening the purity of single-species food ingredients using droplet digital PCR (ddPCR). The screen should be able to flag suspicious samples for further analysis by meta-barcoding, thereby reducing the number of samples and, eventually, the efforts and costs inherent to a meta-barcoding workflow. Similar approaches have been used to estimate the proportion of meat from different species in meat products [15,16,17,18], fish [19], and products of plant origin [20,21]. However, these publications also rely on the correlation between DNA yield and sample intake, whereas the proposed approach does not. Therefore, by excluding a source of variation, the resulting assay should be more robust.

The core idea of the proposed DNA accounting is that, if a single species product is ‘pure’ and if the species is well characterized (in terms of genome size, ploidy, etc.), the number of target copies measured by quantitative PCR should be identical to the ‘expected’ number of target copies calculated from its fluorometrically measured DNA concentration, within the bounds of measurements uncertainty. As a corollary, deviations from the expected copy number are an indication that the product is not ‘pure’ and may be adulterated (e.g., bulking agents may have been added, leading to fewer targets as part of the DNA will come from the bulking agent).

The assumptions of the method are: (I) the target sequence is highly specific and is not present in any adulterant (or at least not in the same amount of copies), (II) all bulking agents used yield DNA with approximately the same efficiency as the product under investigation. Droplet digital PCR was chosen as a method to ‘count’ the number of target sequences since it allows direct absolute quantification unlike qPCR which relies on standard curves of a reference material to obtain absolute measurements. Moreover, it is less affected by the presence of co-extracted inhibitors due to the dilution effect.

We demonstrated the applicability of the ‘DNA accounting’ approach by analyzing 141 commercial saffron samples taken from the markets of 20 EU Member States. Saffron is derived from the stigma and styles of the saffron crocus (*Crocus sativus*). Being one of the most expensive food ingredients, it is a formidable target for economically motivated adulteration [22]. The proposed screening approach could help to systematically survey the authenticity of the product placed on the market. It is not suitable for grading its quality as other parts of *Crocus sativus* such as petals and stamens, which may be used as bulking agents, cannot be recognised.

## 2. Materials and Methods

### 2.1. Samples

#### 2.1.1. Reference Samples

Plant reference material was obtained from the Meise Botanical Garden (Nieuwelaan 38, 1860 Meise, Belgium), reference plant DNA was obtained from the Kew DNA bank (The Royal Botanic Gardens, Kew, DNA Bank. https://www.kew.org/data/dnaBank/, accessed on 13 September 2019) and from the DNA Bank of the Botanic Garden and Botanical Museum Berlin (BGBM). All DNA samples as well as underlying voucher specimens are deposited at the Botanic Garden and Botanical Museum Berlin (BGBM) and are available via the Global Genome Biodiversity Network (GGBN) [23] and the Global Biodiversity Information Facility (GBIF). Plant material and DNA were provided under the agreement of the Convention on Biological Diversity 1992. Reference samples used in this study include *Arnica montana, Beta vulgaris, Bixa orellana, Buddleja officinalis, Calendula officinalis, Crocus sativus, Crocus speciosus, Crocus vernus, Daucus carota*, and *Hemerocallis fulva*.

#### 2.1.2. Samples for Method Development

Samples used for method development and validation were prepared from fresh and dry materials commercially available through shops and garden centers. Where possible, single plants/fruits were used.

#### 2.1.3. Commercial Saffron Samples

Samples (n = 141) were collected in 20 countries of the European market space at various stages of the supply chain. Samples that were sold in powdered form were extracted as is, while samples consisting of whole stigmata were milled.

#### 2.1.4. Sample Preparation

Fresh materials were thinly sliced (<1 mm) and dried (Memmert model UM500 at 75 C) for 2 h or until dry. Dry and dried materials were mixed/milled using an MM301 Ball mill (Retsch) for 2 min at 30 Hz using either 10 mL grinding jars and 10 mm beads (for softer and pre-ground materials) or 20 mL grinding jars and 20 mm beads (harder materials such as seeds). Sample blends were prepared from single species; their mixing ratio (weight/weight percentage) was determined by a microbalance (PG503-S, Mettler Toledo, OH, USA).

### 2.2. DNA Extraction

#### 2.2.1. Automation

Automated DNA extraction of approximately 300 mg of each sample was performed using a Tecan Freedom EVO liquid handler with Promega chemicals (CTAB extraction buffer, CLD lysis buffer, Reliaprep Resin, BWA wash buffer), the Promega Purefood protocol, a sample load volume of 350 μL, and an elution volume of 150 μL. Large volume DNA extraction (>350 mg sample) was performed manually using a CTAB based method adopted from [24].

#### 2.2.2. Quantification

Fluorometric DNA quantification was done on a Qubit 4 fluorometer (Invitrogen™, Thermo Fisher Scientific, Waltham, MA, USA) with High Sensitivity chemistry (Invitrogen) according to manufacturer instructions using 5 μL of sample. For each sample, two independent sample dilutions were quantified twice (two independent standard curves), thus yielding four measurements per sample that were averaged to obtain the DNA concentration estimate.

### 2.3. PCR Primers

#### 2.3.1. Saffron

The *Crocus* genus consists of about 100 recognized species [25] with several closely related to the saffron crocus (*Crocus sativus*). As less expensive species may be substituted or admixed into saffron [26,27], special care was taken to make sure there was no cross-reactivity between crocus species. This was accomplished by aligning sequences for a putative Mg-protoporphyrin monomethyl ester cyclase, a single copy nuclear gene [25], from several crocus species (*C. speciosus*, *C. vernus*, *C. sativus* accession numbers: HE663944.1, HE663941.1, HE663909.1 and HE663908.1 respectively) using clustal Omega [28,29,30]. Primers were picked manually and evaluated using Bisearch [31,32] and in silico PCR [33]. The primer sequences are given in Table 1 (primers ‘Crosat’ and ‘CarthaJQ’).

*Crocus sativus* has a haploid genome weight of 5.9 pg and a ploidy of 3 resulting in a monoploid genome weight of 7.87 pg [34] or approximately 127 genome copies per ng of template DNA. However, the results from the digital PCR indicated that the target sequence is only present on two of the three genome copies (2/3 copies per monoploid genome). Saffron is known to be a probable progeny of *C. cartwrightianus*, which contributes to two of the three genomes, while the other parental lineages remain unclear [35]. As far as we understand it, the targeted sequence is only located on those two *C. cartwrightianus* derived genomes and not on the third one.

The CroSat assay has an estimated resolution of 1.6 under standard conditions (λ = 0.7, approximately 100 ng DNA per reactions) but improves markedly when less than 50 ng DNA per reaction is used; the assay had approximately 0.5% rain (see [36] for details on the performance parameters).

#### 2.3.2. Safflower

*Carthamus tinctorius*: The CarthaJQ primers target a sequence-characterized amplified region (SCAR). SCAR-primers are based on sequenced random amplified polymorphic DNA (RAPD) fragments that show differential amplification between the species of interest. Primers have been adopted from [27]; their sequence is listed in Table 1.

#### 2.3.3. Specificity

Both primer pairs were tested for cross-reactivity with other species by qPCR using SYBRgreen chemistry. The species tested were all published as possible adulterants for Saffron: *Beta vulgaris, Buddleja officinalis, Capsicum annuum, Crocus speciosus, Crocus vernus, Calendula officinalis, Carthamus tinctorius, Daucus carota, Zea mays, Hemerocallis fulva*, and *Triticum aestivum* [26,27,37,38,39]. For the results of the specificity tests, see Appendix A.

### 2.4. PCR Methods

#### 2.4.1. Real Time PCR

Reactions were performed in 25 μL using primers from Table 1 ordered from Invitrogen (standard desalted primers). Reactions were run using PowerUp™ SYBR™ Green Master Mix (Thermo Fisher Scientific, Waltham, MA, USA) and Nuclease free water (Thermo Fisher Scientific). Primer concentration was 200 nM final. DNA template input was 18 ng per reaction, unless otherwise mentioned. All reactions were amplified in ABI microamp 96-well 0.1 mL Fast plates using an Applied Biosystems Quantstudio S7 (Thermo Fisher Scientific). A single thermal cycling protocol was used for all real time PCR reactions: 10 min 95 ${}^{\circ }$C, 60× (15 s 95 ${}^{\circ }$C, 1 min 60 ${}^{\circ }$C). Results were analyzed and exported using the SDS software.

#### 2.4.2. Droplet Digital PCR

Reactions were performed using the Biorad QX200 platform using Twin.Tec 96 well PCR plates (Eppendorf, Hamburg, Germany). Initial volume of the reaction mixture was 20 μL which, together with the droplet generating oil, resulted in a final PCR volume of approximately 45 μL. Reactions were set up using Evagreen Supermix (Biorad, Hercules, CA, USA), primers ordered from Invitrogen, and Nuclease free water (Ambion). Primer concentration was 200 nM final for CarthaJQ and 300 nM for CroSat. DNA template input varied from 5 to 15 ng per reaction depending on the concentration of the DNA extract. Thermal cycling was performed on a ABI Veriti using the following thermal cycling protocol: 10 min 95 ${}^{\circ }$C, 45× (15 s 95 ${}^{\circ }$C, 1 min 60 ${}^{\circ }$C), 10 min 98 ${}^{\circ }$C. Results were analyzed and exported using the Quantasoft 1.6.6.320 software (Biorad, Hercules, CA, USA).

### 2.5. Sequencing and Metabarcoding

#### 2.5.1. Barcode PCR Amplification

The five barcodes recommended by the Consortium for the Barcode of Life (CBOL) Plant Working group [40] were used for metabarcoding. Since the five barcodes have different annealing temperatures, five separated PCR reactions were performed. Usually, 40 ng of DNA were used in each reaction. The barcodes, the primers, and the annealing temperatures are reported in Table 2.

PCR reactions were performed in a volume of 50 μL with primers obtained from Invitrogen and using Gold 360 Mastermix (Thermo Fisher Scientific), DMSO (Merck, Kenilworth, NJ, USA), and Nuclease free water (Ambion). Thermal cycling was performed on a GeneAmp PCR system 9700 (Applied Biosystems) using the following thermal cycling protocol: 10 min 95 ${}^{\circ }$C, 35× (30 s 95 ${}^{\circ }$C, 30 s (temperature see Table 2), 40 s 72 ${}^{\circ }$C), 7 min 72 ${}^{\circ }$C. PCR products were separated by agarose gel electrophoresis and then purified using a column based PCR purification kit (PureLink PCR Purification Kit, Invitrogen) and quantified by fluorescence measurements (Qubit). These products were used as starting material to prepare the DNA barcode libraries for NGS.

#### 2.5.2. Library Preparation and Sequencing

The libraries were prepared using the Ion Plus Fragment Library Kit (Thermo Fisher Scientific), following the manufacturer’s recommendations [41]. All the libraries were evaluated for their quality on the Agilent 2100 Bioanalyzer instrument. Subsequently, the libraries were pooled in an equimolar amount into the template reaction for attachment of the fragments to Ion Sphere Particles (ISP) and clonal amplification in emulsion-PCR. The template reaction was conducted on the Ion OneTouch 2 instrument (Thermo Fisher). Next, recovery and enrichment were performed. Enriched samples were subsequently sequenced on the Ion GeneStudio S5 System (Thermo Fisher), using the Ion 520 chip, which produced 3–5 million reads (1–2 Gb).

### 2.6. Data Processing

#### 2.6.1. DNA Accounting Data Analysis

All calculations and curve fitting were done using R [42] version 3.5.2 (20 December 2018) ’Eggshell Igloo’. The data were exported from the droplet reader as ‘csv files’ and imported into R. Droplet calling was done using the approach presented in [36] using the ‘cloudy’ algorithm version 3.03 as retrieved from Github (https://github.com/Gromgorgel/ddPCR, accessed on 12 February 2021). Non-NGS related sequence analysis (e.g., for primer design and local alignments) was performed in R using functions available through Bioconductor [43] and the ‘DNR’ package available through Github (http://www.github.com/Gromgorgel/R_Scripts, accessed on 12 February 2021).

#### 2.6.2. NGS Data Analysis

The sequencing data obtained were analyzed on the Torrent Suite Software and then with a custom-tailored software for species identification, provided by Thermo Fisher. The software clustered all the reads and then BLASTed against the NCBI nt database (downloaded locally), providing as results the number of reads attributed to a species with a certain degree of identity (by default higher than 99%). In this way, a list of the species detected in each sample was obtained. The results were then analyzed to evaluate how many reads were attributed to the species of interest and how many reads to possible contaminants or adulterants.

## 3. Results

### 3.1. Method Development

The principle of the proposed DNA accounting is based on linear regression analysis to establish a relationship between the measured number of target copies and the expected number of copies derived from the DNA amount and the haploid genome weight of *Crocus sativus*. This relationship was initially established using DNA extracted from *Crocus sativus* bulbs as reference samples of known identity and purity (n = 72, analysed in duplicate). In a later stage, this was repeated with DNA extracted from stigmata of *Crocus sativus* market samples (four samples, analyzed in duplicate) with purity confirmed by NGS (*Crocus sativus* read percentage above 95%). On both occasions, a large number of *Crocus sativus* samples spanning a wide range of copy numbers (independent extracts, independent dilutions) was evaluated using the CroSat assay. These ‘measured copies’ (see Equation (Equation 1)) were regressed against the ‘expected copies’ ($c_{e}$, as calculated from the measured DNA concentration, see Equation (Equation 2)) as shown in Figure 1.
(1)$$c_{m}=\frac{\lambda \times V_{s}}{V_{p}}$$

Here, λ is the average number of target copies per partition as measured by digital PCR, $V_{p}$ is the partition volume (0.00085 μL for the QX200 platform), and $V_{s}$ is the sample volume (20 μL).
(2)$$c_{e}=\frac{DNA\times 1000\times n\times x}{2\times 1C}$$

Here, DNA is the amount (in ng) of template in the reaction, *n* is the number of target copies per haploid genome, *x* is the ploidy of the species, and 1C its monoploid genome weight in pg.

Figure 1 shows a clear correlation between the expected and measured copies with a slope close to one (1.138). However, the residuals plot (not shown) shows a non-constant variance for the residuals (heteroskedasticity). As a consequence the estimated standard errors are inconsistent. This can be overcome by using weighted least squares (WLS) to compute a prediction interval (i.e., the range of measurements in which 95% of all future observations are expected to lie) as shown in Figure 1, provided the samples have comparable purity of the reference sample. However, it should be noted that without explicitly formulating a model for the variance, the results obtained are not mathematically exact but approximate.

Samples whose measurements fall within the prediction interval are thus comparable to the reference samples and can be considered ‘pure’. For samples outside the prediction interval, their purity can be expressed as a percentage (referred to as ‘digital PCR purity’ or ‘percent ddPCR purity’) by the ratio between a sample’s measured result ($c_{m}$) and relevant prediction bound. This way, the percentage reflects how far outside the prediction interval a measurement lies.

Or more formally: let *s* be a sample that has $c_{m,s}$ copies estimated by ddPCR and $c_{e,s}$ expected copies then its percentage dPCR purity (${\%}_{s}$) is given by
(3)$${\%}_{s}=\{\begin{array}{cc} \frac{c_{m,s}}{l_{Ce,s}} & \mathrm{for}\;c_{m,s}<l_{Ce,s}\\ 1 & \mathrm{if}\;l_{Ce,s}\geq c_{m,s}\leq u_{Ce,s}\\ \frac{c_{m,s}}{l_{Ce,s}} & \mathrm{for}\;c_{m,s}>u_{Ce,s} \end{array}$$
where $l_{Ce,s}$ and $u_{Ce,s}$ are the respective lower and upper bounds of the prediction interval at $c_{e,s}$.

### 3.2. Method Validation

Automated DNA extraction worked well for all samples with an average yield of 42.03 ng per μL. The A${}_{260}$/A${}_{280}$ ratio, as checked by UV, was on average 1.864 (>1.8) indicating extracts of sufficient purity for downstream analysis. For a random subset of samples, DNA integrity was checked using automated electrophoresis (Agilent Tapestation, Genomic DNA screentape) and was found to be 4.2 (DIN value) on average.

DNA was extracted from *Crocus sativus* bulbs during method development and for construction of the calibration curve for economical reasons. To ensure the results obtained with DNA from bulbs are compatible with results obtained from stigmas, the ddPCR results of the commercial samples with the highest percentage of NGS reads attributed to *Crocus sativus* (i.e., most certainly pure saffron) were plotted together with DNA extracted from bulbs and the resulting calibration curve is shown in Figure 1. DNA extracted from saffron stigmata and from bulbs behaved in a similar manner, which confirmed that DNA from pure bulbs and stigma can be used interchangeably.

An initial evaluation of the DNA accounting strategy was performed using laboratory prepared samples which consisted of *Crocus sativus* material admixed with various known adulterants (Safflower, wheat flour, and *Capsicum annuum)* in a range of *w*/*w* percentages (5, 10, 20, and 30%). Figure 1 shows where these samples fall with respect to the calibration curve and acceptance range around it. These results show that the sensitivity of the approach differs depending on how well DNA is extracted from the adulterant relative to saffron. For materials that have equal or better extractability (e.g., safflower), the ddPCR-% drops fast when increasing the admixture level (61, 57, 17, and 8% ddPCR purity for 5, 10, 20, and 30% admixture respectively). Whereas materials that yield less DNA per mg have a slower decline of ddPCR-% (75, 68, 55, and 53% ddPCR purity for 5, 10, 20, and 30% admixture of *Capsicum* respectively). In general, the samples with 5% admixture tended to fall close to, but outside, the acceptance range. We therefore estimate that 5% *w*/*w* admixture is about the limit of detection for non-declared biomaterial by DNA accounting, provided the DNA of the admixed material is not difficult to extract.

### 3.3. DNA Accounting of Saffron Samples

For each sample the number of copies of the genetic marker was counted using ddPCR and compared to the expected number of copies, as calculated from the amount of template DNA input and interpreted with respect to a calibration curve constructed from DNA obtained from *Crocus sativus* bulbs. Of the 141 commercial saffron samples, 108 samples were categorized as ‘non suspicious’ (between 80% and 120% ddPCR purity) and 33 were categorized as ‘suspicious’ (less than 80% or more than 120% ddPCR purity). Figure 2 shows a visual overview of these results. To provide a thorough evaluation of the DNA accounting approach, a total of 103 samples were analysed by NGS. A final overview of all samples that were deemed adulterated can be found in Table 3, whereas a complete overview of results is presented in appendix.

All samples categorized as ‘non suspicious’ were found to be botanically pure by NGS metabarcoding (see below). For the ‘suspicious’ samples that had a relatively high percent ddPCR purity (80–70%, >120%), reads were mostly attributed to *Crocus sativus* with a few samples having up to 15% reads attributed to other *Crocus* species but without indication of the presence of other botanicals. Five of the fourteen ‘suspicious’ samples with low ddPCR purity (<70%) also had suspiciously low *C. sativus* read counts (see Table 3), while the *C. sativus* read counts of the remaining samples ranged from 87.5 to 95.5%. This proves that the false-negative rate of DNA accounting is very low: none of the samples containing relevant amounts of non-declared botanicals as found by NGS were classified as ‘non suspicous’ by ddPCR. This highlights the advantage of the described screening approach, which justifies excluding samples categorized as ‘non suspicious’ based on their ddPCR purity from NGS analysis, resulting in a significant reduction of NGS workload (here: 33 instead instead of 141 samples).

To further simplify the data evaluation step, a decision rule was built by combining Equations (Equation 1) and (Equation 2) and omitting the constant terms. This produced a ratio of the measured copy number of a saffron sample and the corresponding amount (in ng) of template DNA in the reaction ($\frac{c_{m}}{DNA}$). The distribution of these ratios represents the variation present within the market (see Figure 3). The ratios were approximately normally distributed with an arithmetic mean value of 143 and standard deviation of 40; thus 95% of the population lies within the boundaries of 65–220 copies/ng (mean ±1.96× standard deviation). New samples can be evaluated by simply checking if their ratio falls within this range; if so, they can be considered as being uncontaminated with non-declared botanicals.

However, the use of the aforementioned strategy should not be done without at least an in-house verification using several authentic samples, as it is strongly dependent on the results obtained from DNA quantification.

### 3.4. NGS Metabarcoding

In the present study an average of 8261 reads per sample was obtained. The NGS software grouped reads into clusters if at least three identical reads were present; clusters with less than three reads were disregarded. Overall, none of the samples analyzed resulted in 100% of the reads attributed solely to Saffron (i.e., *Crocus sativus*), in samples that fell within the prediction interval, the average read % for *Crocus sativus* was 91.5%. The bulk of the remaining reads was attributed to ‘*Crocus* spp.’ without exact species returned (on average 8.7%). Samples within the acceptance range (but outside the prediction interval) had a an average read % for *Crocus sativus* of 89.0%. The majority of samples outside the acceptance range had similar read % (i.e., 90.3%, outliers removed) with the exception of the samples listed in Table 3 whose average is 7.7%. Those five samples had particularly low saffron read counts (see Table 3) and are considered adulterated. In four of these samples *Carthamus tinctorius* (safflower), a known saffron adulterant [26,27,39] was detected with significant read counts (57%, 90%, 88%, 80%) and its presence was confirmed using a safflower specific qPCR [27].

A full breakdown of the results can be found in supplemental material.

### 3.5. qPCR Confirmation of Results

For the samples in which NGS analysis indicated the presence of safflower, qPCR confirmation of its presence was performed by amplifying a specific target for *Carthamus tinctorius* (Table 1). All four reactions showed amplification of the target and the amplicon had the same melting temperature as a known reference sample (DNA extracted from *Carthamus tinctorius* seed material). These results confirmed the adulteration of these samples as already indicated by both the sequencing and DNA accounting analysis.

## 4. Discussion

DNA based methods have become a widely used tool for detecting fraud in the agri-food chain, particularly for species identification and quantification of certain food ingredient [6,7,8,9]. Species-specific PCR is the method of choice to target a particular adulterant, e.g., horse meat in beef patties, which, by multiplexing, can be extended to target several, known adulterants. Another popular approach, Sanger sequencing of barcoding regions, while efficient for identifying species, does not allow targeting several non-declared species (adulterants) at the same time.

Authentication of culinary herbs and spices goes beyond the question whether the named species, e.g., *Crocus sativus*, is present but tries to answer how much of the named species makes up the sample, in other words how ‘pure’ the sample is. The challenge in using quantitative PCR is the transformation of DNA amount (mass or number of copies) into mass (fraction) of the biological material. In its simplest format, binary mixtures (weight/weight) of the adulterant/named species are prepared, DNA is extracted, and after real-time PCR the cycle threshold (Cq) values are plotted against the log-transformed mixture percentage. Including a ubiquitous reference gene makes the assay more robust and compensates to a certain degree the effect of processing on DNA extractability and integrity [44]. However, most published PCR assays aim at quantifying an adulterant using primers specific for the non-declared species, whereas for assessing ‘purity’ the named species itself has to be quantified.

By assuming that the DNA content is proportional to the mass of biomaterials, Rong Chen et al. [45] developed a real-time PCR assay for estimating directly the mass of saffron in saffron containing herbal products; furthermore, they showed that the amount of DNA was fairly constant for several different batches of saffron.

We have taken this idea one step further and suggest a broadly applicable technique for estimating the ‘purity’ of biological matter using digital PCR. This technique is less prone to interference by amplification inhibitors and allows the estimation of the number of DNA copies present in a sample without external references. By measuring the number of genome copies in a sample and comparing it to the number of copies calculated from the known amount of DNA used in the reaction, ‘purity’ of a biomaterial can be estimated, provided the species of interest is well characterized (genome size, ploidy, target copy number). The number of target copies measured should be identical to the ‘expected’ number of target copies as calculated from its DNA concentration (measured flourometrically) within the bounds of measurements uncertainty.

When regressing the expected against the measured copy numbers over a wide range of saffron samples, the slope of the regression line was 1.138 (Figure 1), proving the assumption that ‘purity’ of a biomaterial can be assessed by dPCR. However, the estimated slope was not fully in line with the ideal value of 1.00. The reason for this could be inherent to the instrument used, as Low et al. [46] also found deviations from the ideal value when comparing measured copy numbers to dilutions of a certified reference material (ERM-AD623), which were attributed to the particular brand of ddPCR instrument. Furthermore, inaccuracies in the analytical chain are inevitable: fluorometric quantification of the DNA in the extract, the 1C values used, DNA damage during extraction, PCR efficiency, etc. which may all contribute to the deviation of the slope from unity.

We have applied the technique to the analysis of 141 commercial samples of saffron taken in 20 EU Member States. It has to be stressed that the ‘purity’ estimate obtained by ddPCR is intended for screening purposes and not to draw firm conclusion whether non-declared botanicals are indeed present. For confirming this, samples flagged as suspicious were analysed using NGS metabarcoding as a means of compositional analysis. If NGS indicated species known to be potential adulterants, their presence was confirmed by qPCR of a species-specific target.

ddPCR screening indicated more samples as suspicious than could be confirmed by NGS. This was partially expected as the DNA used for the construction of the reference curve for DNA accounting was obtained from *Crocus sativus* bulbs, and although precautions have been taken to minimize difference in DNA-quality between reference and samples (i.e., the bulbs were cut, dried, and milled), it stands to reason that the variability of DNA quality, and therefore of testing results, is higher amongst samples. Differences in processing, drying, milling, storage, microbial decontamination (e.g., gamma irradiation [47]), etc. may adversely affect both DNA quantification and its specific target amplifiability.

DNA accounting has proven to be a valuable analytical approach in the food fraud detection workflow. The ability to rapidly screen large numbers of samples for their purity is a much needed capacity in today’s food chain with its rapid turn-over. The suggested approach has significant value as a screening tool and has also a potential for a confirmatory assay to estimate the purity of botanicals. For the latter approach, the inclusion of a suitable and ubiquitously present reference gene in the assay would be necessary to compensate for variations in DNA extractability and amplifiability.

## Figures and Tables

**Figure 1 foods-10-02670-f001:**
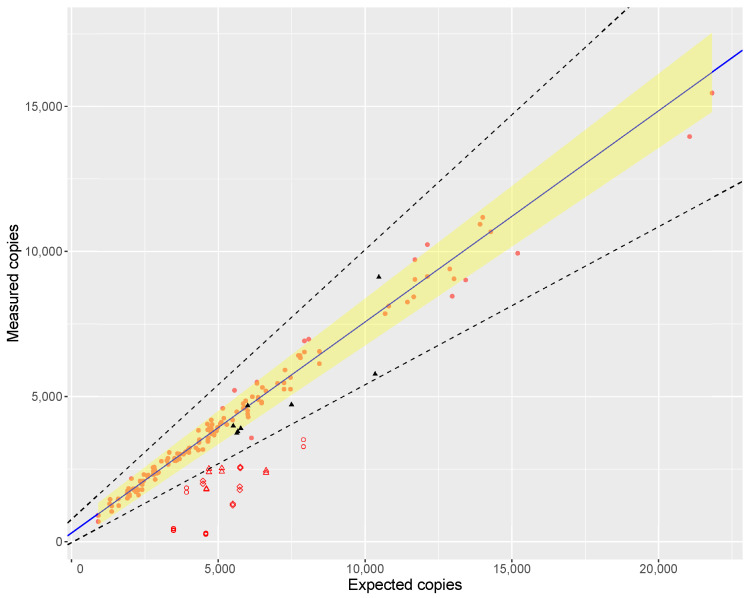
Correlation between measured and expected target copies. The figure shows the *Crocus sativus* reference samples (●), their correlation line (blue), and 95% prediction interval (shaded area). The boundaries of the acceptable range (80% and 120% ddPCR purity) are shown as dashed lines. Results for the stigma samples are shown in black (▲) and laboratory prepared admixtures are shown in red (∘ for samples amdmixed with safflower, ⋄ for samples admixed with wheat flour, for samples admixed with *Capsicum annuum*).

**Figure 2 foods-10-02670-f002:**
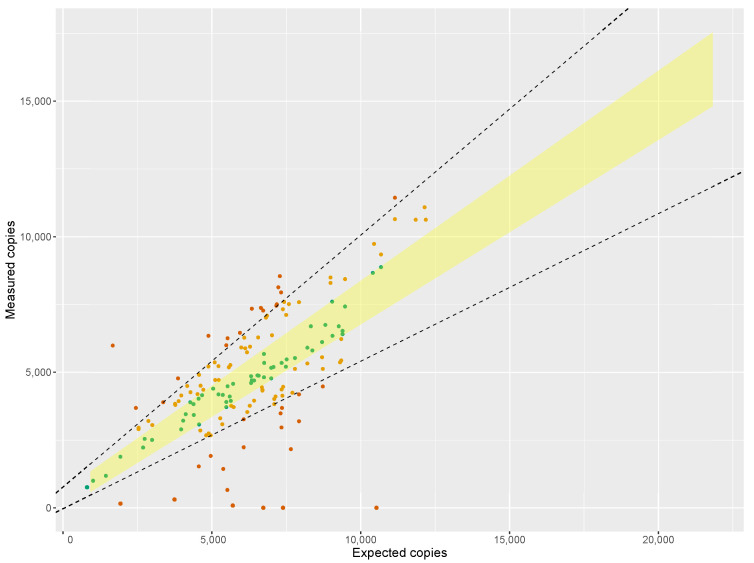
Overview of the market sample results: measured copies per sample plotted in function of their ’expected copies’, the shaded area represents the 95% prediction area based on the regression of the reference samples. The boundaries of the acceptable range (80% and 120% ddPCR purity) are shown as dashed lines. Samples have been colored depending on their categorization: green for 100% ddPCR purity, orange for 80–120% ddPCR purity, red for <80% or >120%.

**Figure 3 foods-10-02670-f003:**
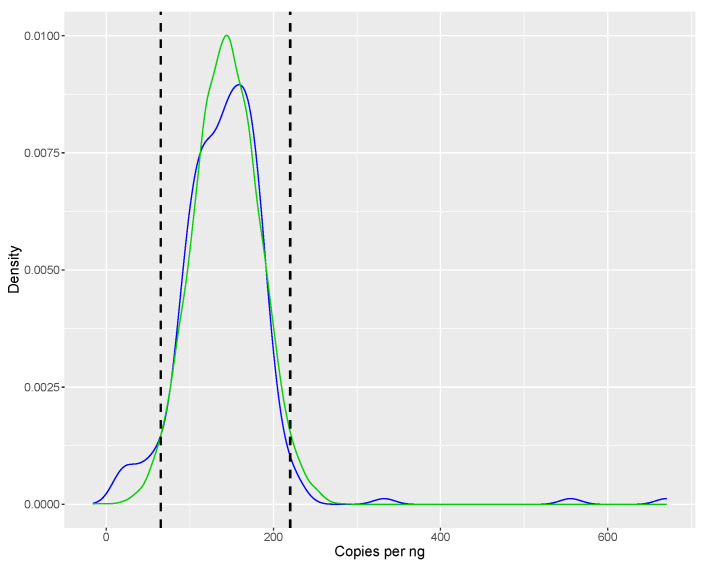
Density plot of the number of copies per ng template as measured by digital PCR in samples of high purity as confirmed by NGS. Dotted vertical lines indicate the boundaries between which 95% of the samples can be found. In green a normal approximation of the distribution is shown.

**Table 1 foods-10-02670-t001:** Sequences and amplicon lengths of primers used in this study. Amplicon length for Crosat and CarthaJQ was estimated from the sequences with Genbank accession number HE663909 and JQ952667 respectively.

Name	Forward (5′-3′)	Reverse (5′-3′)	Length
CroSat	GAACTGGTGTCAGGATGAGA	GGCCATGAATTAATGATGCAA	153
CarthaJQ	ACAACCATTGGAGATTCCGG	AGTGAGCACTCTTAGTTAACC	131

**Table 2 foods-10-02670-t002:** List of barcodes with primer sequences, annealing temperatures, and the mean expected amplicon sizes.

Barcode Name	Primer Name	Sequence (5′-3′)	Annealing Temp	Amplicon (bp)
RbcL	rbcL-a-F	ATGTCACCACAAACAGAGACTAAAGC	55 ${}^{\circ }$C	560
	rbcL-a-R	GTAAAATCAAGTCCACCRCG		
TrnL	trnL(UAA)-c	CGAAATCGGTAGACGCTACG	50 ${}^{\circ }$C	500
	trnL(UAA)-d	GGGGATAGAGGGACTTGAAC		
psbA	psbA-trnH –F	GTTATGCATGAACGTAATGCTC	64 ${}^{\circ }$C	430
	psbA-trnH-R	CGCGCATGGTGGATTCACAATCC		
MatK	matK-1RKIM-F	ACCCAGTCCATCTGGAAATCTTGGTTC	52 ${}^{\circ }$C	800
	matK-3FKIM-R	CGTACAGTACTTTTGTGTTTACGAG		
ITS	ITS2-F	ATGCGATACTTGGTGTGAAT	56 ${}^{\circ }$C	460
	ITS2-R	GACGCTTCTCCAGACTACAAT		

**Table 3 foods-10-02670-t003:** Overview of the samples found suspicious by DNA analysis. For each sample the table shows its percentage digital PCR purity, the % of reads attributed to *Crocus sativus* in NGS metabarcoding, the contaminants as identified by NGS, and whether or not the contaminant presence has been confirmed by qPCR markers specific for that contaminant.

ID	% dPCR Purtity	Read % (NGS)	Contaminant (NGS)	qPCR Confirmed
SH00906	0.09	0.02	*Carthamus tinctorius* (safflower)	Yes
SH00917	13.18	38.39	*Carthamus tinctorius* (safflower)	Yes
SH00947	0.04	0.00	*Carthamus tinctorius* (safflower)	Yes
SH00740	2.21	0.01	*Tegetes* spp. (marigolds)	No
SH00199	0.06	0.03	*Carthamus tinctorius* (safflower)	Yes

## Data Availability

Data is available in the Appendix. For raw data disclosure please contact the authors.

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
