# Peer review of "DNA Accounting: Tallying Genomes to Detect Adulterated Saffron"

_foods, 2021, doi:10.3390/foods10112670_

Round 1
Reviewer 1 Report
The manuscript describes the development of a quantitative assay, based on digital PCR, to evaluate the potential adulteration of saffron, with other plant material used as a bulking agent.
The procedure is rigorous, conclusions are sound and the method is promising as a preliminary screening for the identification of commercial samples suspicious of fraudulent admixtures, despite inner limits that are clearly reported
However, some points should be better clarified and missing information should be added, as reported below:
Page 2 -Other non-targeted methods developed for plant species identification in food, based on DNA profiling should be mentioned, as Tubulin Based Polymorphism and SpinDel.
Page 4 -
- PCR primers: pPlease specify to which species the aligned sequences belong
- sativus has a haploid genome weight of 5.9 pg and a monoploid genome of 3.93. is it correct??
- The description of the CroSat assay parameters in dPCR (resolution, rain value..) should be more clearly explained for readers not familiar with dPCR
Page 7 - “Samples whise measurement …” I believe I’s a typo
Page 8 and figure 2B - what is the digital purity of the samples plotted in Fig 2B? Maybe these few samples and related values from NGS could be added in Table 4 for a useful comparison to suspicious samples.
Page 8 - admixed samples. No information is given about these samples. In order to better understand results obtained from admixed samples, it is important to know from which starting material they have been prepared and the relative w/w proportions. A Table listing the admixed samples reported in Fig 1B and the percentage admixture should be added.
Page 9 - “DNA extracted from C. sativus bulbs…..” the verb is missing in this sentence
Table 4 – was the presence of marigolds in sample SH00740 tested by Tegetes specific primers? If so, qPCR should be mentioned in paragraph 3.5 and primer sequences should be reported in the Methods section
Figure 2: Please indicate what different sample colors and the dotted lines indicate.
Page 10 - NGS metabarcoding. Did the authors also run a NGS on a pure saffron sample as a control? It would be useful to see if a 100% attribution of reads can be reached
Table A1: I warmly suggest to add %s values or, at least, categorization by dPCR for the reported samples, in order to check correspondence with NGS results
All Figures: Axis legends should be enlarged to be more clearly readable
Reviewer 2 Report
This paper described a digital PCR-based DNA accounting method for adulterated saffron detection.
The results are interesting, and the method seems to be robust. I recommend addressing the following problems.
1. The presentation of the manuscript needs to be improved. For instance, I recommend to draw a table of contents to describe the significance and the key points of the paper.
2. The figures seems to be raw data. Larger fonts are needed.
3. Measured copies and expected copies may be very different for a non-fraud product, according to the accuracy of digital PCR. How to avoid mistakes in detection?
4. Is the method applicable to practical applications? it seems to be time consuming and labor extensive.
Reviewer 3 Report
This is an apparently interesting study since it investigates the application of the ‘DNA accounting’ approach by analyzing commercial saffron samples from EU market. The methodology is well described, and the results are presented with clarity and detail.
Author Response
Reviewer 3 did not ask for changes to the document. As such there are no answers to provide.